# Analysis of Risk Factors for African Swine Fever in Lombardy to Identify Pig Holdings and Areas Most at Risk of Introduction in Order to Plan Preventive Measures

**DOI:** 10.3390/pathogens9121077

**Published:** 2020-12-21

**Authors:** Silvia Bellini, Alessandra Scaburri, Marco Tironi, Stefania Calò

**Affiliations:** Istituto Zooprofilattico Sperimentale della Lombardia ed Emilia Romagna, Via Bianchi 9, 25124 Brescia, Italy; alessandra.scaburri@izsler.it (A.S.); marco.tironi@izsler.it (M.T.); stefania.calo@izsler.it (S.C.)

**Keywords:** African swine fever, pigs, surveillance

## Abstract

In 2019, the area of the European Union (EU) affected by African swine fever (ASF) expanded progressively in a southwestern direction from Baltic and eastern countries. The disease can severely affect and disrupt regional and international trade of pigs and pork products with serious socioeconomic damages to the pig industry. Lombardy is one of the most important European pig producers and the introduction of ASF into the pig population could adversely affect the entire sector. A study was carried out to identify the farms and territories in the region most at risk of ASF introduction to plan preventive measures. The pig holdings were identified through a descriptive analysis of pig movements and Social Network Analysis (SNA), while, for the identification of the most exposed municipalities, an assessment of risk factors was carried out using the ranking of summed scores attributed to the Z-score. From the analysis, it was found that 109 municipalities and 297 pig holdings of the region were potentially more at risk, and these holdings were selected for target surveillance. This information was provided to veterinary authority to target surveillance in pig farms, in order to early detect a possible incursion of ASF and prevent its spread.

## 1. Introduction

ASF is a serious viral disease of domestic and wild pigs caused by African swine fever virus (ASFV) which is currently representing a major threat concerning the swine industry worldwide. Trade restrictions accompanying the occurrence of the disease can disrupt regional and international trade of animals and animal products with a serious economic impact on the swine sector [1]. The number of countries or territories reporting the disease presence has increased in the last few years and ASF has officially been notified to the Word Organization for Animal Heath (OIE) by member countries from sub-Saharan Africa, Europe and Asia [1]. This situation is expected to worsen with currently ASFV-free pig-producing countries being at increased risk of ASFV incursion due to globalization, increased transportation of people and animal products as well as illegal trade and importation of potentially infected pigs and pig products [2].

Over the past decade, ASF has spilled out of its original area of endemicity and has spread widely in Europe and Asia. The number of countries or territories affected by ASF has increased in sub-Saharan Africa, Asia and Europe [3]. In 2007, the disease was reported in trans-Caucasian countries and has spread ever since to the Russian Federation and several Eastern European countries. In 2019, the area of the EU affected by ASF expanded progressively, moving mainly in a southwestern direction from Baltic and eastern countries [4].

Italy is one of most important European producers of pigs [5] with a presence in Lombardy of 50% of national pig population [6]. Moreover, about 1.8 million of pigs are introduced annually into the region to meet the needs of the processing industry. The Lombard pig sector is of particular economic relevance for the entire country also in consideration of the processing industry, which is aimed at the production of high-quality pork products [6,7]. Therefore, the introduction of an epidemic disease into the pig population could adversely affect the entire productive sector.

In the current study, the risk factors affecting the spread of ASF in domestic pigs were considered. The movement of living animals and of the means of transport represent the main risk for the spread of diseases and this is particularly relevant in areas with high stocking density. Incursion of ASFV in a previously free territory is often occurring through the introduction of contaminated pig products (swill) or living animals into pig farms, while the spread of the disease within a country happens most likely due to poor farming conditions [1,2,3,4]. Therefore, an understanding of the patterns of animal contact networks provides essential information for the representation of risk-based surveillance and also to establish disease control strategies. Indeed, the premises with a high number of outgoing movements are key players for the disease spread, whereas, premises with a high number of incoming movements are likely to be at risk of disease introduction.

Recent analysis carried out by European Food Safety Authority (EFSA) identified a group of potential risk factors that could play a role in the spreading of ASF in domestic pigs. In the infected territories, swill feeding, the presence of free-ranging pigs and home slaughtering, as well as the presence of a substantial number of smallholders in the area, were found significant indicators for the spread of ASF in the domestic pig sector [4]. The high percentage of smallholders is an important indicator of the potential spread of ASF in the territory due to poor biosecurity conditions and other features that are typical of this type of farming practice such as swill feeding, illegal animal movements, collections of backyard pigs in markets and home slaughtering [4]. In some countries, free-ranging pig farming is a traditional husbandry practice in which pigs are generally raised in extensive or semiextensive production systems where contact with wild pigs is facilitated. In fact, the presence of free-ranging pigs in certain EU territories is another element that has contributed to the spread and maintenance of the virus in these territories [8]. In smallholdings and in free-ranging pigs, there is often a low level of compliance with legal requirements on registration and identification of pigs and these deficiencies reduce the effectiveness of disease management, hampering the eradication process [4]. Wild boars are susceptible to ASF and in the current EU situation; they have contributed significantly to the maintenance and spread of the virus. Furthermore, several field observations about the possible contribution of infected wild boars to the spread of ASFV to domestic pigs have been reported [9].

The present study did not consider soft ticks as a possible mean for ASF transmission in the region. Indeed, ticks of the genus Ornithodoros have a significant role in ASFV transmission in Africa. The ecological niche of these ticks in Europe has not been adequately determined. However, ticks do not play a significant role in the geographical spread of the virus while in endemic situation they serve as a reservoir allowing the virus to persist locally in the environment [10]. Lombardy is ASF free, this implies that the scenario just described does not reflect the current epidemiological situation of the region.

The aim of the study is to identify the geographical areas of the region where risk factors for the spread of the disease are present and to identify pig farms that are central to the commercial network and that, therefore, can play a role in the spread of the disease. This information is relevant for the veterinary authorities in order to design a risk-based surveillance and to plan targeted control activities.

## 2. Results

### 2.1. Introduction of Pigs from Other Italian Regions and within Lombardy

In 2018, 1,779,797 pigs from 291 farms outside the region were introduced to Lombardy. These animals were introduced through 3947 commercial batches in 741 Lombard holdings. In the analysis, the number of farms from which pigs were purchased, the number of animals and the number of batch of pigs introduced into a farm were considered factors increasing the likelihood of pathogens being introduced into the farm.

The Lombard holdings that purchased pigs from more than three farms outside the region are considered to be more exposed to the risk of introducing ASF. 

In Figure 1 is reported the distribution of holdings, which purchased pigs in other regions. On the left side of the graph, is observed that 500 Lombard holdings purchased 808,612 pigs from only one farm outside the region, by moving 1941 batches of pigs while on the right side, is shown one Lombard holding that purchased 14,665 pigs from eight different holdings through 29 batches of pigs.

The analysis showed that 76.3% of pigs introduced to Lombard farms come from pig movements within the region. In fact, 3225 Lombardy holdings introduced 5,737,556 pigs from 823 holdings of the same region. 

The Lombard holdings that purchased pigs from more than four farms were considered to be more exposed to the risk of introducing ASF. 

In Figure 2 is reported the distribution of holdings, which purchased pigs in Lombardy. On the left side of the graph, it is observed that 2468 Lombard holdings purchased 2,244,696 pigs from only one farm, by moving 7838 batches of pigs. On the right side, is shown a Lombard holding that purchased 17,909 pigs from 10 different holdings through 38 batches of pigs.

### 2.2. Social Network Analysis

In Figure 3a,b, we report the graphical representation of our Social Network Analysis.

Figure 3a shows the movements of pigs to Lombardy from other regions. The nodes (holdings) number 1032 and the links (movements) 1117; the colors represent the productive typology of the holdings. In Figure 3a, some areas with a dense and closed network have been highlighted, circled in blue. 

The Figure 3b shows pig movements within the region. The nodes are 3514 and the links are 4523. The network was too complex to interpret graphically SNA. 

Table 1 shows the results of SNA indicators. In both networks, the density was near to 0, indicating that nodes were isolated and trading relationships were in general unique. The diameter of the within region network was 40, meaning that the two holdings that were more distant were linked through 40 farms, whereas the diameter of the network of the movements from other regions was 22. The range of in-degree centrality within the region was from 0 to 154 farms, which means that the maximum of holdings of the region had purchased pigs from the same holding inside the region was 154. The range of out-degree centrality from other regions was from zero to eight; this indicates that the maximum number of different holdings of other regions that sold pigs to the same farm in Lombardy was eight, as already pointed out in Figure 1. The range of out-degree centrality within region was from zero to 10; this indicates that the maximum of different Lombard holdings which sold pigs to the same farm in Lombardy was 10, as already pointed out in Figure 2.

The results of closeness centrality and of betweenness centrality confirmed that holdings were not closed and in general, no farm had an important role as intermediary in either network.

### 2.3. Identification of the Municipalities More Exposed to the Risk of Introducing ASF

In Lombardy, 109 (7.2%) municipalities resulted more exposed to ASF: 54 of these are classified as medium and 55 as high risk (Table 2).

The wild boars are present in 611 municipalities (40.5%), 15 of which resulted at medium risk and 11 at high risk for ASF.

The analysis showed that, in Lombardy, there were 58 farms with wild boar. In 40 of these, there were only wild boar, while in 18 there were both wild boars and domestic pigs. Moreover, 55.2% of these wild boar farms were classified in the regional database (BDR) as noncommercial pig holdings, 32.8% as commercial farms and 12.0% as outdoor farms.

From the study on pig movements, carried out by SNA and the threshold of the 95th percentile, 57 Lombard farms resulted more exposed to the risk of spreading pathogens both for movements of pigs within the region and for movements from other regions. These farms were mainly located in the provinces of Brescia, Mantua and Cremona. Farms classified at higher risk of transmission due to pig movements within the region and located in municipalities with medium and high risk are 35 while those considered at high risk due to pigs introduced into the region and located in municipalities with medium and high risk are 22 (Figure 4).

Based on the results of the analysis, as a first step, the following holdings were selected to target ASF surveillance in domestic pigs:Fifty-eight farms of wild boar or mixed domestic pigs—wild boar, 10 of these were located in medium or high-risk municipalities, 48 in low-risk municipalities;Eighty-eight holdings more exposed on the basis of the introduction of pigs from other regions, 22 were located in municipalities classified as medium or high risk;One hundred and twenty-eight holdings more exposed based on pig movements within the region, 35 of these were located in municipalities classified as medium or high risk;Dealer’s premises were zero in Lombardy but 8 holdings were identified for target surveillance because they purchased pigs from dealer’s premises located in the neighboring regions;Fifteen holdings were identified to target surveillance because in National Database (BDN) were registered as non-commercial farms but the analysis of the movements showed that they sold pigs to other non-commercial farms.

Overall, the holdings that were central to the trading network were selected for target surveillance, priority of control was given to the holding located in municipality at risk of ASF spread. The analysis was carried out for each farm of the region, considering also certain factors that may be relevant for the transmission of pathogens such as the number of commercial partners, the number of animals introduced as well as the number of batches of pigs introduced into the holdings, by giving control priority to holdings located in the areas identified as having the higher risk of ASF spread. In fact, the control activities should be proportionate to the risk level of the holding taking into consideration that the receiving holdings are more at risk of introducing the disease, while the holdings with outgoing movements can play an important role in its spread.

## 3. Discussion

The results of the study showed that the majority (81.06% = 16,892/20,893) of pig batches movements in Lombardy occurred within the region and, as expected, breeding holdings had a pivotal role in the commercial network of the region, they sold 3,589,561 pigs to fattening holdings and 199,497 pigs to breeding operations.

However, the results of the analysis showed that in the productive orientation “fattening” are registered holdings that buy and sell pigs (also to breeding holdings) and those that only buy pigs. In the regional trading network a fattening holdings was identified who sold pigs to both breeding (99,517 pigs) and fattening farms (1,834,216 pigs). This means that this farm could play a central role in the secondary spread of ASFV within the pig farms of the region and therefore this shall be considered when farms are selected for surveillance. Indeed, enhancing surveillance and biosecurity could have a significant impact in terms of minimizing the extent of a potential epidemic. Following the indication reported by the European Commission [11], the official controls on the identified holdings shall be focused on the correct registration and identification of pigs and pig holdings, the verification of the health status of the animals (examination of pig plus sampling, in case of clinical signs suggestive of ASF) and verification of the correct implementation of farm biosecurity principles. Target surveillance is performed to complement passive surveillance, which remains the most effective tool to detect new ASF cases at an early stage in previously disease-free areas. Indeed, passive surveillance is foreseen throughout the country and provides for the collection of samples for laboratory investigation in case of clinical signs resembling ASF and when ante or post mortem signs raise suspicion of the disease.

The analysis of pig movements carried out in this study was based on data for only one year and this is a limitation of the study, in terms of generalization of the findings. However, the results of this study made an important contribution to the design of risk-based surveillance in the regional ASF control program for the year that was about to begin. The study is continuing with the analysis of pig movements in the following years, this will make it possible to have more solid and comprehensive results and understand whether there is a seasonal trend in pig movements in Lombardy as well, particularly in certain subpopulations of pigs (backyards). This would make it possible to better establish, even temporally, the surveillance activities. Indeed, intensifying surveillance during certain periods could be more effective for disease prevention and control. Underreporting of pig movements by the owners could be another limitation of the study, however this is difficult to quantify due to the lack of data.

## 4. Materials and Methods 

### 4.1. Description of the Area: Lombardy Region

Lombardy is a region of the Northern part of Italy where intensive livestock husbandry and pig production is one of the most important sectors in livestock production. The region has a surface of 23,863.1 km^2^ divided into 11 provinces and 1507 municipalities. Based on the data recorded in the BDR, the pig population is about 4.3 million in 8887 holdings, which represents 49.9% of the national pig population. Regional pig density is 181.5 animals/km^2^. The provinces in which pig farming is more intensive are Brescia (2225 farms, 1,319,476 pigs), Mantua (744 farms, 1,093,470 pigs) and Cremona (519 farms, 895,548 pigs) (Figure 5). The province of Brescia is the most populated, and in some areas pig density is above 275.7 animals/km^2^. 

In Italy, detailed information on the size of the wild boar population is lacking. However, according to a very rough estimate carried out by the Italian Institute for Environmental Protection and Research (ISPRA), in the country there are no less than 600,000 wild boars [12]. In Lombardy, based on data provided by the Regional Veterinary Service (RVS), there is an estimated number of wild boars of about 12,000 animals. At present, a more precise indication on the number of heads is not available and, for the purpose of the study, which was conducted at a municipality level, the presence or absence of wild boar was considered.

### 4.2. Data

#### 4.2.1. Source of Data

Data on pig movements were provided by the RVS, whereas, the data referring to the characteristics of pig holdings located in Lombardy were extracted from the BDR. Information about pig holdings located in other Italian regions which moved pigs to Lombardy was extracted from the BDN.

For the purposes of this study, movements of pigs from farms to other farms (“movements for life”) were considered, while we have excluded from the analysis the movements of pigs that have been moved directly from farms to abattoirs. 

#### 4.2.2. Data Collection

In Italy, livestock movement registration is mandatory. In Lombardy, pig movements are registered in the BDR. The movement record is defined as the movement of batch of pigs between two holdings. The records relating to the movements of pigs occurred in 2018 were extracted. Each movement record consisted of the date of the movement, the departure holding, its location, the holding of destination, the number of pigs moved and the geographical position of the holding of destination.

The characteristics of pig holdings, such as their typology (holding, genetic center and dealer’s premise), productive orientation (fattening, breeding, non-commercial farms), productive technique (open cycle, closed cycle, weaning, growing, finishing), localization (region, province, municipality and geographic coordinates) and number of pigs present in holding, were collected from the BDN and BDR. 

For the identification of the municipalities more exposed to the risk of ASF introduction, the holdings were grouped as: commercial farms, outdoor pig farms and noncommercial farms, following the indications reported on DG SANTE working document [11], which is the strategic approach for the management of ASF in the EU member states. Data on the presence and number of holdings of wild boars in the municipalities under study were provided by the RVS and they were referring to the 2016–2017 period. 

### 4.3. Statistical Analysis

#### 4.3.1. Introduction of Pigs from Other Italian Regions and within Lombardy

The data of the movements to other farms (for life) and for the slaughterhouse were grouped by production type of the farm of origin and by geographical position and were calculated: the number of farms, the number of animals introduced and the number of movements carried out. The results were represented by a graph: in blue the number of farms that introduced pigs to Lombardy, in orange the number of animals introduced and in black the number of pig batches. For the purposes of the study and to establish a first priority of control, the Lombard holdings that purchased pigs from more than three farms outside and more than four farms inside the region were considered to be more exposed to the risk of introducing ASF. These farms are the ones that in the distribution were above the threshold fixed to the 95th percentile.

#### 4.3.2. Social Network Analysis

For the analysis on pig movements, data were divided into two groups: (1) data on the movement of pigs from other regions (introduction of pigs into Lombardy), (2) data on pig movements within the region.

The SNA [13] was used to analyze the trading patterns of pigs in Lombardy and to identify the holdings that are central in the trading network of the region. The units of interest, nodes, were pig holdings, the links between nodes represent the commercial relationship of pigs between holdings, the arrows showed the direction of the movements, from seller to buyer. All movements involving the same couple of operations were considered as a single link. The output of the SNA was assessed both graphically and analyzing the indicators; in particular, density and diameter were used to describe the network and three centrality measures were calculated to evaluate the strength of the links [14]. The definitions of SNA parameters are reported in Table 3. 

#### 4.3.3. Identification of the Municipalities More Exposed to the Risk of ASF Introduction

To identify the municipalities more exposed to the risk of introducing ASF, the analysis of certain risk factors, such as: (a) presence/absence of wild boars, (b) density of noncommercial pig holdings, (c) pig density of out-door farms and (d) pig density of commercial farms was carried out. Moreover, in this evaluation were also considered the results obtained from the analysis of pig movements in Lombardy, pig introductions in Lombardy and pig movements within the region (e). 

For each municipality, the density values for each of the four risk factors (b–e) were calculated based on the geographical position of each farm. 

To compare risk factors, Z-scores were used. In particular, the Z-scores of density (Z) were computed with the following formula:$$Z=\frac{X-\mu }{\sigma }$$
where X was the density of the risk factor of a Lombard municipality, μ and σ were the mean and the standard deviation density of the Lombard municipalities, respectively. 

Finally, each z-score was ranked as follows: Z < 2, score = 0; 2 ≤ Z < 3, score = 1; Z ≥ 3, score = 2. 

Based on the different sum of scores of the risk factors, the municipalities were ranked into three risk classes: high (summed scores > 1, presence of at least one risk factors), medium (summed score = 1, presence of one risk factor) and low (summed score = 0, no risk factor) [15].

#### 4.3.4. Software

SNA was performed using library “igraph” [16] of statistical software R version 3.6.1 [17]. Indicators of SNA were calculated with software UCINET [18]. The maps were created with ArcGIS 10.2 (ESRI, Redlands, CA, USA). 

## 5. Conclusions

In over 40 years of infection, the presence of ASF in Sardinia has entailed a limited risk for the rest of the Italian territory, while the development of the ASF situation in Eastern European countries is causing particular concern in the disease-free countries, including Italy. The EU strategy for the management of ASF has been developed to establish harmonized measures in response to the ASF situation [11]. It is addressed to all member states who should adopt a risk-based approach considering their epidemiological situation and in ASF-free territories. The main activities should be focused on ASF prevention and early detection. For this reason, given the risk that the ASF could pose to the pig industry in Lombardy, a study was conducted to identify territories and pig holdings more exposed to the risk of introducing the disease, in order to plan control measures. 

The movement of pigs is one of the main risks for the spread of diseases and results show that in Lombardy, 81% of the pig movements for life (to other pig holdings) occurred within the region and this is relevant to consider for the secondary spread of the disease, in case of ASF introduction.

The fact that the holdings pinpointed for surveillance fall into risk municipalities or not, provided an additional indication to establish priority of control. 

Data recording in BDN is essential to establish surveillance activities but, in order to be effective they should be updated and reflect correctly the commercial attitude of the holding, which implies its potential role in spreading pathogens. The analysis of pig movements conducted in Lombardy, revealed certain discrepancies between the production orientations recorded in BDN and the actual business activity of certain holdings. This shortcoming should be properly addressed considering their actual commercial activity. Indeed, the wrong classification of the holdings can influence the effectiveness of the official control plans whose measures are normally established taking into consideration the production orientation of the holdings, which also implies their risk of spreading pathogens.

## Figures and Tables

**Figure 1 pathogens-09-01077-f001:**
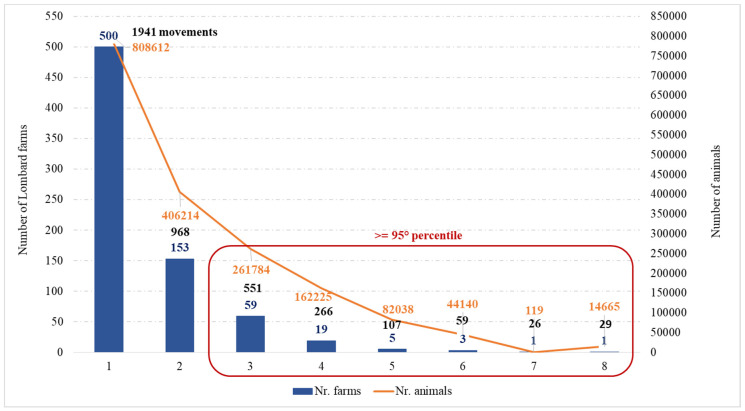
Distribution of holdings introducing pigs from other Italian regions. The number of farms that introduced pigs to Lombardy (in blue), the number of animals introduced (in orange) and the number of pig batches (in black). The X-axis represents the number of different holdings from which a Lombard farm purchased. The principal Y-axis represents the number of Lombard farms which purchased. The secondary Y-axis represents the number of pigs.

**Figure 2 pathogens-09-01077-f002:**
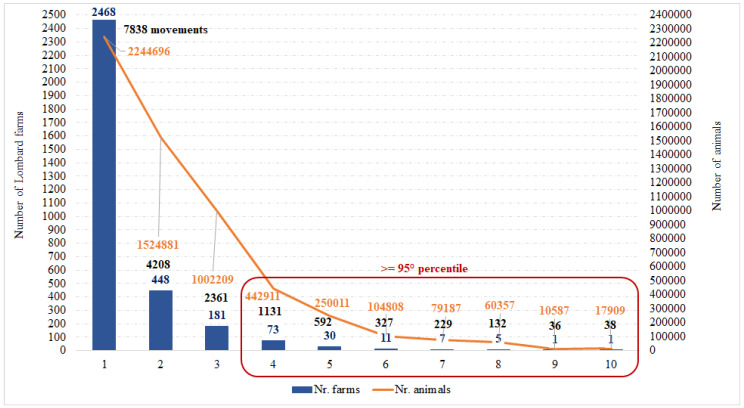
Distribution of holdings introducing pigs from holdings of the region. The number of farms that introduced pigs to Lombardy (in blue), the number of animals introduced (in orange) and the number of pig batches (in black). The X-axis represents the number of different holdings from which a Lombard farm purchased. The principal Y-axis represents the number of Lombard farms which purchased. The secondary Y-axis represents the number of pigs.

**Figure 3 pathogens-09-01077-f003:**
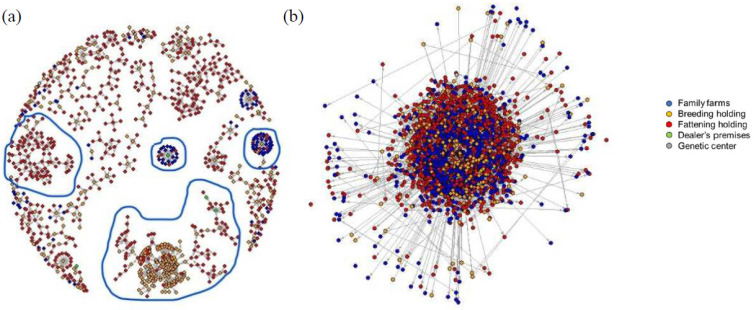
Social Network Analysis (SNA) of movements: (**a**) from other Italian regions to Lombardy, (**b**) within Lombardy.

**Figure 4 pathogens-09-01077-f004:**
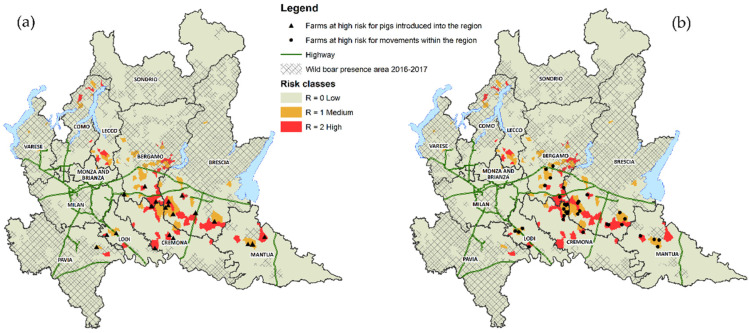
Maps of municipalities for risk classes with farms at high risk due to movements (**a**) from other regions (**b**) within the region.

**Figure 5 pathogens-09-01077-f005:**
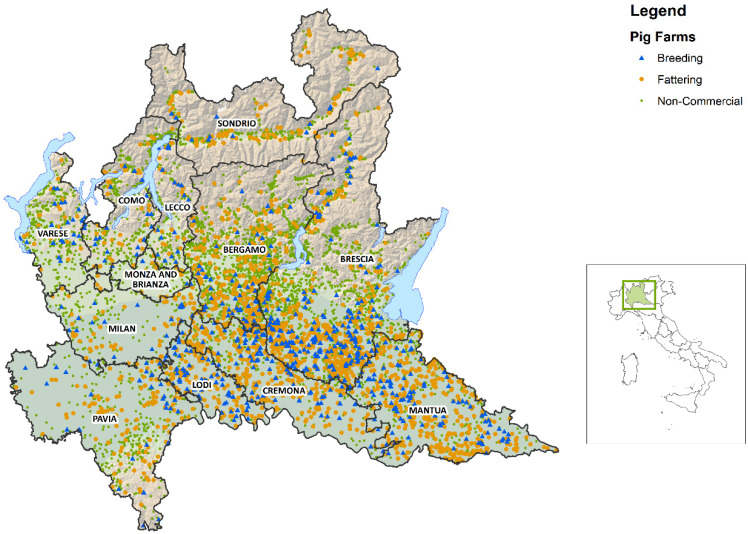
Distribution of pig holdings in Lombardy.

**Table 1 pathogens-09-01077-t001:** Results of SNA parameters of within region network and from other region network.

SNA Parameters	Within Region	From Other Regions
Density (n)	0.001	0.001
Geodesic distance (mean)	15.803	8.886
Diameter (n)	40	22
Degree centrality (*)	0.073 (0–4.412)	0.209 (0–4.656)
In-degree centrality (*)	1.286 (0–154)	1.079 (0–47)
Out-degree centrality (*)	1.286 (0–10)	1.079 (0–8)
Closeness centrality (*)	0.340 (0.337–0.342)	0.339 (0.336–0.340)
In-closeness centrality (*)	0.042 (0.028–0.333)	0.106 (0.097–0.277)
Out-closeness centrality (*)	0.030 (0.028–0.031)	0.101 (0.097–0.107)
Betweenness centrality (*)	0.115 (0–14.115)	0.222 (0–10.647)

* Mean (min—max).

**Table 2 pathogens-09-01077-t002:** African swine fever (ASF) risk distribution of municipalities and presence of wild boar, the risk factors are considered one by one, in the last row all the risk factors are included using the sum of the rank.

Risk Factors	N. Municipalities with Low Risk	With Wild Boar Presence	N. Municipalities Medium Risk	With Wild Boar Presence	N. Municipalities With High Risk	With Wild Boar Presence
Density of pig movements	886	382	51	14	53	11
Density of non-commercial farms	1021	488	48	15	49	11
Density of out-doors farms	36	19	1	0	1	0
Density of commercial farms	682	269	37	8	45	8
Overall risk	1398	585	54	15	55	11

**Table 3 pathogens-09-01077-t003:** Definition of SNA parameters.

SNA Parameters	Definition
Density	Proportion of links among all possible network links. Range from 0 (all nodes are isolated) to 1 (all nodes are connected).
Pathway	Single path between two nodes.
Geodesic distance	The number of relations in the shortest possible path from one node to another.
Diameter	The largest geodesic distance in the network.
Degree centrality	The number of links of each node.
In-degree centrality	The number of farms from which each farm receives animals.
Out-degree centrality	The number of farms to which each farm sends animals.
Closeness centrality	How many paths are required for a particular node to access every other node in the network
In-closeness centrality	The number of paths from which each farm receives animals.
Out-closeness centrality	The number of paths to which each farm sends animals.
Betweenness centrality	The number of shortest paths between all other nodes that go through a particular node. It measures the importance of a particular node as an intermediary in the network.

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
