# Peer review of "Analysis of Risk Factors for African Swine Fever in Lombardy to Identify Pig Holdings and Areas Most at Risk of Introduction in Order to Plan Preventive Measures"

_pathogens, 2020, doi:10.3390/pathogens9121077_

Round 1

Reviewer 1 Report

This study is interesting  as it seeks to predict the risk of a potential introduction of a veterinary disease that could have an impact on the economic sector.
However, the methodology has some flawes.
Major remarks
- What is the impact of wild boar on the risk of introducing of ASV? Indeed, as stated, their objective is to measure the risk of introduction through the exchange of animals (pigs) between Lombardy and other regions. As long as wild boars are not included in this exchange, they should not be included in the risk analysis. If this is the case, they should also take into account the presence of ticks, potential vectors of the disease.
- In the methodology, it is stated that for the identification of the risk of introduction, the companies were grouped into commercialfarms, out-door farms and non-commercial farms. However, in the analysis of the results obtained, the impact of each type of farm in the risk of ASV introduction is not clear. 
In the results presented in section 2.2 (social network analysis) and in figures 3a and 3b, it is difficult to understand the impact of these types of farms in the introduction of the disease, in addition, other parameters are added to the results (family farms; breeding, fattening...).
Minor remarks
-In the description of the study area, the authors state that the provinces of Brescia, Mantua and Cremona are the areas of intensive pig farming. However, in the analysis and presentation of the results, it is impossible to know if these areas are at high risk of disease introduction.

- It would be great to give the total number of farm in the Lombardie region
- In the introduction section, it would be good to give additional information on the level of risk of the disease instead of just saying that it is a major threat (line 25-29).
- Lines 40 and 41 can be removed from the manuscript. They are redundant with the end of the introduction (lines 73-75).
- In the results section, in the lines (84-86), the last two sentences should be moved to the methodology section.
-In the methodology section, section 4.3 should not be entitled "methodology".
-The conclusion must be rewritten. The way it is presented does not tell what conclusions can be drawn from this study. It is a repetition of the objectives of the study and the background.

Reviewer 2 Report

In the given study Authors described risk analysis of probable introduction of one of the most dangerous and devastating diseases of domestic pigs and wild boars – African swine fever into Lombardy, a strategic pig-holding-region of Italy. For more than 10 years the disease has been affecting east Europe and recently spread to Asia and Western Europe - increasing its range and posing the treat or neighboring countries. The disease leads to enormous economic losses therefore study provided by Authors should be considered as important and justified.

Introduction is well written, interesting and sufficient, however the results need some clarification.

There is not clearly presented how the results showed in the Table 1 implicate overall risk estimation (is the SNA analysis was also ranked?), also the ranked risk estimation presented in the Table 2 is not clearly revealed how the ranking was done (probably basing on Z-score analysis but X parameter should be revealed – how the “density of risk factor” was estimated). Maybe for the better understanding, the of example of raw data and analysis should be implemented as supplementary material.

In the lines 131-132 we may read that 55 municipalities were assigned to those with low risk, in the headline of Table 2 we may read that these municipalities were assigned to high risk group, while the lowest risk municipalities (headline of Table 2) are presented with the highest rank.

Minor issues:

Line 34-39: Please, if it is possible add specific references for the data in the paragraph.

From the text we know that the X-axis of Figure 1 represents the number of farms that provide the pigs for Lombardy holdings, however it is not well explained in the Figure - please add the description.

Please add the description to X-axis on the figure 2.

In conclusion, the presented article is interesting but some issues need clarification i.e. explain the risk estimation processes for the readers. 
